

# Influence of CNTRENE® C100LM carbon nanotube material on the growth and regulation of *Escherichia coli*

Brittany Twibell[1], Kalie Somerville[1], Geoffrey Manani[2], Molly Duszynski[2], Adam Wanekaya[2] and Paul Schweiger[1]

[1] Biology Department, Missouri State University, Springfield, MO, United States of America
[2] Chemistry Department, Missouri State University, Springfield, MO, United States of America

## ABSTRACT

The growing use of carbon nanotubes (CNTs) in industrial and consumer products raises important questions about their environmental fate and impact on prokaryotes. In the environment, CNTs are exposed to a variety of conditions (e.g., UV light) that could lead to decomposition and changes in their chemical properties. Therefore, the potential cytotoxic effect of both pristine and artificially aged carboxyl functionalized CNTRENE® C100LM CNTmaterial at neutral and acidic conditions on *Escherichia coli* K12 was analyzed using a minimal inhibitory concentration (MIC) assay, which also allowed monitoring of non-lethal growth effects. However, there were no observable MIC or significant changes in growth behavior in *E. coli* K12 when exposed to pristine or aged CNTs. Exposure to pristine CNTRENE® C100LM CNT material did not appear to influence cell morphology or damage the cells when examined by electron microscopy. In addition, RNA sequencing revealed no observable regulatory changes in typical stress response pathways. This is surprising considering that previous studies have claimed high cytotoxicity of CNTs, including carboxyl functionalized single-walled CNTs, and suggest that other factors such as trace heavy metals or other impurities are likely responsible for many of the previously reported cytotoxicity in *E. coli* and possibly other microorganisms.

## INTRODUCTION

Carbon nanotubes (CNTs) are a type of nanoparticle with the potential for many technological applications, but many CNTs have unknown cytotoxicity. CNTs are cylinders of various lengths composed of single layers of carbon, called graphene, and can be single-walled (SWCNTs) or multi-walled (MWCNTs). CNTs can be modified with functional groups, increasing their potential applications by allowing them to bind macromolecules (*Kolosnjaj-Tabi, Szwarc & Moussa, 2012*; *Chen, Xie & Yu, 2011*). Consequently, the industrial and commercial usage of CNTs has increased several fold over the past decade, and the research and development of new products incorporating CNT materials is rapidly growing. CNTs are used as additives in composite materials,

Corresponding author
Paul Schweiger,
pschweiger@missouristate.edu

such as CNT resins, that are used for a variety of products from wind turbine blades to sporting good equipment (*De Volder et al., 2013*). CNTs have also been used as additives in different types of coatings and films, such as protective paints containing MWCNTs, used in the marine industry and solar cells (*Köhler et al., 2008*; *De Volder et al., 2013*). Recent development of flexible touch screen displays that include SWCNTs have the potential to replace traditional indium tin oxide coated displays, and 60% of cell phone and tablet devices on the consumer market already use lithium ion batteries containing CNTs (*De Volder et al., 2013*; *Köhler et al., 2008*). There is also interest in the use of CNTs in biosensors and drug delivery systems based on functionalization (*De Volder et al., 2013*).

Despite the broad applications of CNTs, many questions remain about environmental and biological safety because materials at the nanoscale have physiochemical properties that differ from their bulk material (*Environmental Protection Agency, 2010*; *Reinhart et al., 2010*). As of 2013 the EPA has implemented a Significant New Use Rule (SNUR) under the Toxic Substance Control Act which specifically refers to CNTs (*Environmental Protection Agency, 2008*; *Environmental Protection Agency, 2015*). This SNUR allows the EPA to track and review chemicals before manufacturing or importing and make decisions based on potential impacts to humans and ecosystems (*Environmental Protection Agency, 2015*), demonstrating the need for further research on the impact CNTs. Of particular interest is lifecycle analysis, which looks at impacts of a chemical and potential release points from production of the raw materials, use in products, end of life recycling and disposal methods, and waste produced at any step in the life cycle (*Köhler et al., 2008*; *Environmental Protection Agency, 2010*). As part of a lifecycle analysis it is important to consider various environmental conditions that a chemical may experience. When materials are deposited in the environment they are exposed to weathering processes that can be mimicked with the use of a UV accelerated weathering chamber (*Grujicic et al., 2003*). UV-light exposure has been shown to cause physical changes in CNT shape and chemical changes, including changes in the way oxygen associates with the CNT wall surfaces (*Grujicic et al., 2003*). The alterations in physiochemical properties of these aged CNTs leads to the question of whether environmental induced changes could affect cytotoxicity of CNTs. However, no studies on the effects of aged carboxyl functionalized CNTs on bacterial cytotoxicity have been done to date. The effect of CNTs on the growth and viability on the bacterial community is a key part of an environmental life cycle assessment of chemicals as bacteria are important factors in nutrient cycling and community structure.

The toxicity of various CNTs has been studied, yet research is often contradictory. This is often due to insufficient characterization of the nanoparticles. During synthesis, carbon sources are used along with metal catalysts, such as cobalt, yttrium, iron, and nickel, which can lead to metal impurities in raw CNTs and often contributes to or enhances toxicity (*Johnston et al., 2010*; *Köhler et al., 2008*; *Kolosnjaj-Tabi, Szwarc & Moussa, 2012*; *Puretzky et al., 2000*). Small variations in the physiochemical characteristics and the type and level of metal contamination of the CNT can influence cytotoxicity, leading to conflicting results in toxicity studies (*Horie et al., 2012*). Physiochemical differences resulting in conflicting toxicity studies highlights the importance of evaluating the impact of CNTs. The growth behavior, metabolism, and gene regulation of the model organism *Escherichia coli* K12

is well established, making it a common first choice microbe for cytotoxicology studies. In this study, the ability of pristine and artificially aged carboxyl functionalized CNTs to inhibit *E. coli* K12 growth was examined using a range of commercial Brewer Science® CNTRENE® C100LM carbon nanotube material (hereafter referred to as CNTRENE material). Gene expression of *E. coli* K12 exposed to CNTRENE material was evaluated and compared to native gene expression by RNA sequencing. These results provide insights into the microbiological safety of this commercially available CNTRENE® product currently used in advanced memory devices for computers, tablets, smart phones, and digital cameras.

## MATERIALS AND METHODS

### Bacterial growth and media

*Escherichia coli* K12 strain SMG 123 (ATCC PTA-7555) was grown in lysogeny broth (LB) or M9 minimal salts medium with the addition of 1 mM thiamine and 2% glucose (hereafter M9 medium) at 37 °C and 200 rpm in a Thermo Scientific MaxQ 400 incubator unless stated otherwise.

### Carbon nanotubes

Pristine carboxyl functionalized CNTRENE® C100LM CNT material, supplied by Brewer Science, Inc., was suspended in distilled water at 135 µg/mL (pH 7.0) and were of the same lot as previously used by *Woodman et al. (2016)*, wherein the physical characterization was described. Briefly, the CNTs had a total metal ion content of less than 25 ppb. Carboxyl functionalization was estimated to be at 2–6% and mainly observed at the open ends of the CNTs. The CNTs had a length range of 0.3–1.5 µm (90% of CNTs) and a diameter range of 0.7–3 nm (95% of CNTs), with the average length of 0.87 µm and width of 1.56 nm. The pristine product was made up of SWCNT, DWCNT, and MWCNT at 70%, 25%, and 5%, respectively.

### Carbon nanotube aging process and spectroscopic characterization

To simulate environmental weathering, pristine CNTRENE material was aged as supplied in distilled water in a QUV Accelerated Weathering chamber (Q-Lab Corp, Cleveland, OH, USA) as previously described (*Woodman et al., 2016*). Briefly, 14 mL pristine CNTRENE material was exposed to 3–4 h alternating ultraviolet and condensation cycles for 12 days. The UV cycle and condensation cycle temperatures were 68 ± 0.5 °C and 47 ± 0.5 °C, respectively. The current of the lamps were 0.5–0.6 amperes and the condensation cooling fan set point was 15. Distilled water was used to provide moisture and served as temperature control bath. The Raman, FTIR, and UV-vis spectra of both pristine and aged CNT material were quantified after sonication for one minute to disperse the CNTs. For Raman spectroscopy, 2–10 µL of the sample was dropped onto an aluminum foil wrapped around a microscope glass slide. The sample was air-dried in a clean environment free from any dust or other contaminants. A Horiba LabRAM HR800 spectrometer equipped with a 50 mW 532 nm excitation laser with a detection capability in the range of 200 to 4,000 nm was used for Raman Photoluminescence spectroscopy at ambient temperature. For FTIR

spectroscopy, 20 µL of the CNTs were put into separate pre-cleaned dry mortars preheated at 80 °C. The sample was dried on the mortar and approximately 400 µg of dry potassium bromide was added and ground into fine powder to make a KBr pellet. The pellet was used for IR measurements using a Bruker IR spectrophotometer. The background data was obtained using a KBr pellet without CNTs. For UV-Vis analysis, CNTs were diluted 10-fold with deionized water and spectra taken with a PerkinElmer Lambda 650 UV-Vis spectrometer at room temperature.

## Minimal Inhibitory Concentration (MIC)

A broth microdilution MIC assay on a 96-well transparent C-bottom plate was performed with a standard inoculum of $5.0 \times 10^5$ CFU/mL in a final reaction volume of 200 µL as previously described (*Wiegand, Hilpert & Hancock, 2008*). The MIC was determined for CNT concentrations from 1.05 µg/ml to $6.44 \times 10^{-5}$ µg/ml and were assayed using a minimum of three replicates from three independent cultures ($n = 9$). Plates were incubated at 37 °C in a BioTek EL808 plate reader with shaking at the medium shake rate and optical density at 595 nm was monitored for 24 h. The Gen5 software (BioTek, Winooski, VT, USA) was used for data collection and growth curves and doubling times were used to evaluate growth effects of CNT exposure.

## Antibacterial plate counts

The effect of pristine CNTRENE material concentrations greater than 5 µg/mL on the growth of *E. coli* K12 was evaluated by the spot-plate technique using a modified method of (*Gaudy Jr, Abu-Niaaj & Gaudy, 1963*). *E. coli* K12 was inoculated to $5.0 \times 10^5$ CFU/ml in 96-well transparent C-bottom plates and grown in LB (pH 7) with addition of pristine CNTRENE material (0 µg/mL, 8.44 µg/mL, 16.88 µg/mL, and 33.75 µg/mL) in a final volume of 200 µL in triplicate. Plates were incubated for 24 h as described above. After incubation, serial ten-fold dilutions of the overnight cultures were performed in 96-well plates and 10 µl of the dilutions were spotted in triplicate on LB plates and incubated at 37 °C for 16–18 h. Colony forming units per milliliter were calculated for each concentration of CNTs and compared to that of the unexposed control group.

## Electron microscopy

Morphological change of bacterial cells exposed to CNTs was evaluated by scanning electron microscopy (SEM) and atomic force microscopy (AFM). *E. coli* K12 was inoculated to a concentration of $5.0 \times 10^5$ CFU/ml in a 200 µl LB culture in a 96-well transparent C-bottom plate containing 0 µg/mL, 33.75 µg/mL, 16.88 µg/mL, and 1.05 µg/mL pristine CNTRENE material. *E. coli* K12 was grown at each CNT concentration in triplicate for 24 h at 37 °C. Replicates were combined, pelleted by centrifugation at 5,000 x g for 5 min, and then washed three times in 0.1 M phosphate buffer (pH 7.2) to remove growth medium. Cells were dehydrated by an ascending ethanol wash series (50%, 70%, 80%, 90%, 95%, 100%, and 100%) with a 5 min exposure to each concentration of ethanol. Samples were transferred onto silicon wafers in 10 µL volumes and either frozen in liquid nitrogen and freeze dried overnight for SEM or allowed to air dry overnight for AFM. SEM was done with a JEOL JSM-7600F field emission scanning electron microscope under vacuum

$(9.6 \times 10^{-5}$ Pascal). SEM images were captured using an accelerating voltage of 1.00 kV and a working distance (WD) between 5.1 mm and 5.2 mm at total magnification ranging from 10,000× to 20,000×. AFM imagines were captured utilizing the Veeco Dimension 3100 with a Nanoscope IIIA Controller under atmospheric conditions $(1.01 \times 10^5$ Pascal). AFM imagines were captured on tapping mode (scan size 5.000 μm or 20.0 μm, scan rate 1.001 Hz, 512 samples) using a silicon tip with a nominal radius of 8.0 nm.

### RNA sequencing

Differential gene expression of CNTRENE material-exposed bacterial cells was evaluated with RNA sequencing. An overnight *E. coli* K12 culture was inoculated 1:100 in 5 mL of M9 medium. Pristine CNTRENE material was added at 1.05 μg/mL to experimental cultures, with all cultures set up in triplicate and incubated to mid-log phase. RNA was extracted using the Qiagen RNeasy Plus Mini Kit (Qiagen, Hilden, Germany), including an on-column DNase treatment (Qiagen RNase-Free DNase Set) according to the manufacturer's instructions.

Extracted RNA was initially quantified using an IMPLEN nanophotometer followed by analysis with an Agilent Technologies 2100 Bioanalyzer system and 2100 Expert software to confirm RNA quality. High integrity samples (RIN > 8) were depleted of ribosomal rRNA using the bacterial Ribo-Zero rRNA Removal Kit (Illumina, San Diego, Ca, USA) according to the manufacturer's instructions. The SMARTer Stranded RNA-Seq kit (Clontech, Mountain View, Ca, USA) was used to prepare cDNA from RNA samples according to the manufacturer's instructions. An Illumina MiSeq instrument (single-end 50 bp read length) was used for RNA sequencing and sequencing data was analyzed using the DNAstar Lasergene Suite Qseq software by the University of Wisconsin- Milwaukee Great Lakes Genomics Center. Sequencing was done from three independent RNA preparations for each sample type. The sequence reads were mapped and analyzed using the RNA-Seq pipeline default parameters using *E. coli* K12 MG1655 as a reference genome. Differential gene expression was analyzed using the student *t*-test with the false discovery rate restricted to 0.05 as the *p*-value adjustment method (*Benjamini & Hochberg, 1995*).

### Data analysis

All statistical analyses were done in using the GraphPad Prism 5.0 software.

## RESULTS

### Effect of aging on CNTRENE® C100LM CNT material

Exposure to environmental conditions is known to change the physiochemical properties of CNTs, which has the potential to influence toxicity (*Valsami-Jones & Lynch, 2015*). To mimic the effect of weathering on CNTs, the pristine CNTRENE material was aged by exposure to UVA at 340 nm using a QUV Accelerated Weathering chamber that has been used previously to simulate outdoor weathering. Accelerated aging of materials in the QUV Accelerated Weathering chamber for between 1,000 h–1,800 h has been equated to a year of Florida summer sunlight exposure (*University of Delaware, CfCM, 2002*) and has been used to investigate the stability of polymers such as polymer-bound hindered

amine light stabilizers (*Macleay, 1989*), to predict the service life of exterior automotive coatings (*Shanbhag, 2012*), the photodegradation of wood-plastic composites (*Peng et al., 2014*), and carbon fiber-reinforced polymer composites (*Tcherbi-Narteh, Hosur & Jeelani, 2013*), among others. The UV-Vis spectra of pristine CNTRENE material had a peak at about 250 nm that is due to the first interband transition of the nanotubes (Fig. 1). As the nanotubes age, the above peak appears to experience a redshift to 270 nm. Additionally, another peak emerges at about 210 nm. This is attributed to the loss of $\pi$-structure of the nanotubes in the aging process. In the Raman spectra, the major observation is the difference in the ratios of characteristic CNT bands (Fig. 2). In general, the aged CNTRENE material exhibited lower D/G and RBM/G band ratios compared to pristine CNTs. The decreasing ratio of bands, especially RBM/G, implies diminishing CNT character as the CNTs age. Previous research on aging of nanocarbons under ambient conditions indicated a decrease in the net structural defects with aging (*Yang et al., 2009*). This is in line with the results observed from the decrease in the intensity of the D band compared to the G band, leading to a decrease in the D/G ratio giving an indication of decreased disorder or defects in the CNTs with aging. From the FTIR spectra, it can be inferred that there is an O–H peak ($\sim$3,430 cm$^{-1}$), C–H stretching as observed in alkanes (and possibly aldehydes or carboxylic acids), and a carbonyl stretch at 1,653 cm$^{-1}$–1,701 cm$^{-1}$ confirmed the presence of a carboxylic acid group (–COOH) (*Chen et al., 1998*; *Liao & Zhang, 2012*) (Fig. 3). Deviations were observed for the intensities of the C =O (with a likelihood of H-bonding as that observed for diketones) and O–H peaks for the aged CNTs being higher than those of pristine CNTs. It can be further inferred that the diketone functionalities are dominant in the aged CNTs compared to the pristine CNTs based on the intensities of the peaks. The band at $\sim$1,653 cm$^{-1}$ may be attributed to C=C stretching (*Chang et al., 2006*). The peaks at 1,385 cm$^{-1}$ and 1,090 cm$^{-1}$ correspond to the expected C–O–H and the C–C–C bending. The peaks at $\sim$2,920 cm$^{-1}$ suggest the presence of a C–H stretch in C–CH$_3$ that can be attributed to protonation of the CNTs as a result of their interactions with water in extreme aging conditions. It is possible that such interactions could occur in the environment because of exposure to favorable reaction conditions such as humidity, presence of hydrogen, oxygen, and heat. The differences in peak intensities suggest probable oxidation of the CNTs with aging, especially evident from the increase in O–H groups. Additionally, aged CNTs had increased coiling and bundling compared to pristine CNTs (Fig. 4). Together these data imply morphological and functional changes as the CNTs are aged in conditions that mimic prolonged environmental exposure.

## Cytotoxicity of CNTRENE® C100LM CNT material exposure

The inhibition and cytotoxicity of an increasing gradient of CNTRENE material up to 1.05 µg/ml on *E. coli* K12 was examined by a MIC assay. Growth curves for *E. coli* K12 in LB (pH 7) in the presence of pristine CNTRENE material over the tested concentration range were very similar to the growth observed in the unexposed control group, with all growth curves overlaying (Fig. 5A). The same trend was observed for *E. coli* K12 in LB (pH 7) in the presence of aged CNTRENE material (Fig. 5B). Cultures grown in LB (pH 7) had an

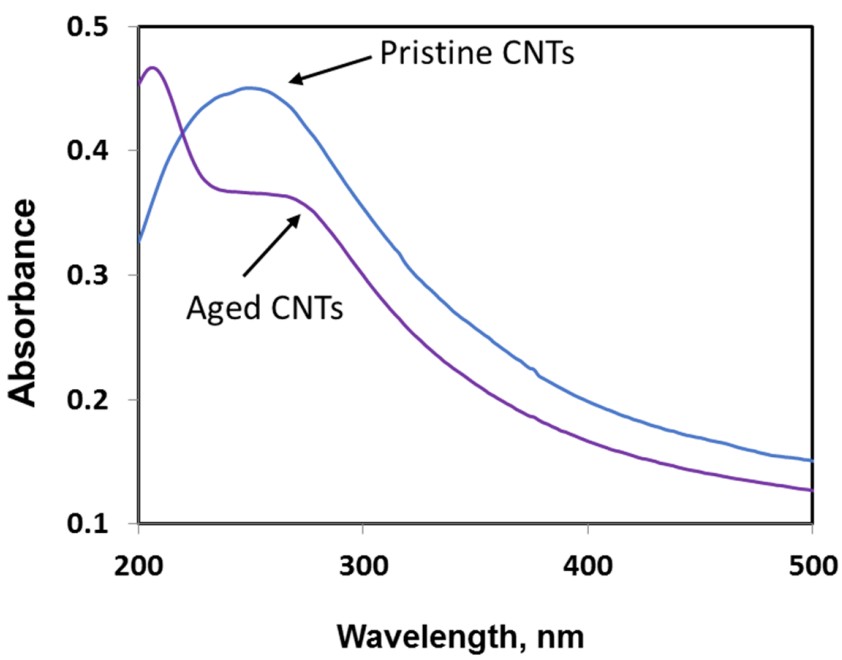

**Figure 1  Comparison of UV-Vis spectra of pristine and aged CNTRENE C100LM CNT material.**

average doubling time of 22.4 min ($\pm$1.2 min) after exposure to pristine CNTs, and an average doubling time of 22.3 min ($\pm$1.1 min) after exposure to aged CNTs (Fig. 6A). In comparison, the unexposed control groups in the pristine and aged CNT assays had doubling times of 22.7 min ($\pm$0.8 min). Doubling times observed for *E. coli* K12 in LB (pH 7) were similar between pristine and aged CNTRENE material concentrations regardless of the concentration used. Indeed, doubling times were not correlated to increasing exposure to either pristine ($r = 0.0995$, $n = 9$ for each concentration, $p = 0.4873$) or aged ($r = 0.2581$, $n = 9$ for each concentration, $p = 0.0675$) CNTRENE material.

The influence of CNTRENE material on *E. coli* growth was analyzed in M9 medium, in which *E. coli* have a slower growth rate and a lower final cell density. In this minimal medium, growth with pristine or aged CNTRENE material was similar to the unexposed control group (Figs. 5C–5D). Unexposed cells had an average doubling time of 60.7 min ($\pm$0.7 min). In comparison, *E. coli* exposed to pristine CNTRENE material had an average doubling time of 60.4 min ($\pm$2.3 min), and when exposed to aged CNTRENE material an average doubling time of 59.9 ($\pm$0.7 min) was observed (Fig. 6B). Considering these data, it is not surprising that there was not a correlation between doubling times and the concentration of pristine ($r = 0.0109$, $n = 9$ for each concentration, $p = 0.9414$) or aged ($r = -0.2688$, $n = 9$ for each concentration, $p = 0.1381$) CNTRENE material exposure.

Lastly, the effect of pH on CNTRENE material toxicity was examined using LB medium at pH of 5. As for the other conditions, the growth of *E. coli* was not inhibited by either pristine or aged CNTRENE material exposure, having growth curves that superimposed (Figs. 5E–5F). For pristine CNTRENE material treatment, an average doubling time of 30.3 min ($\pm$0.8 min) was observed. Similarly, aged CNTRENE material treatment produced an

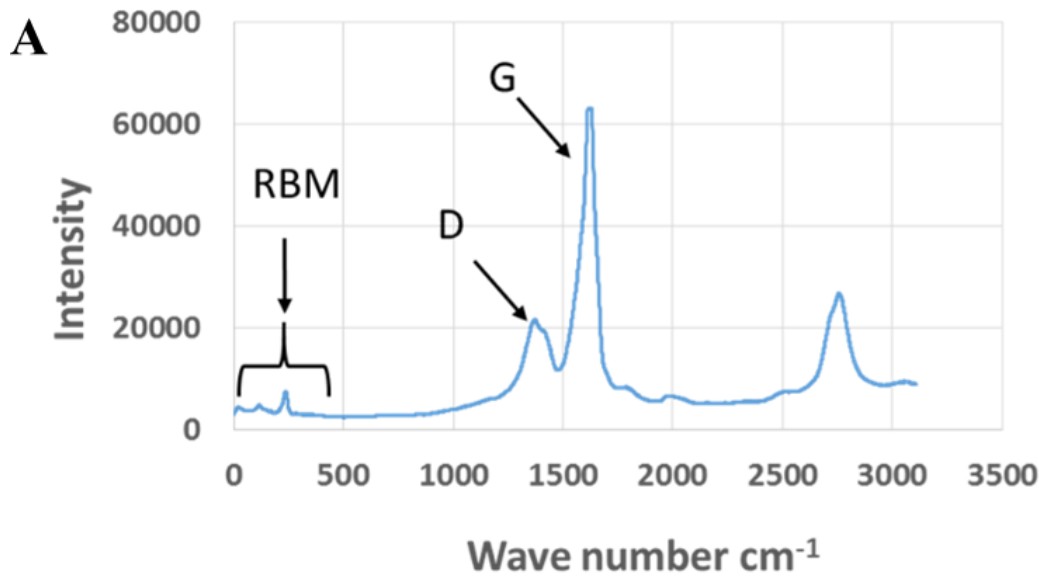

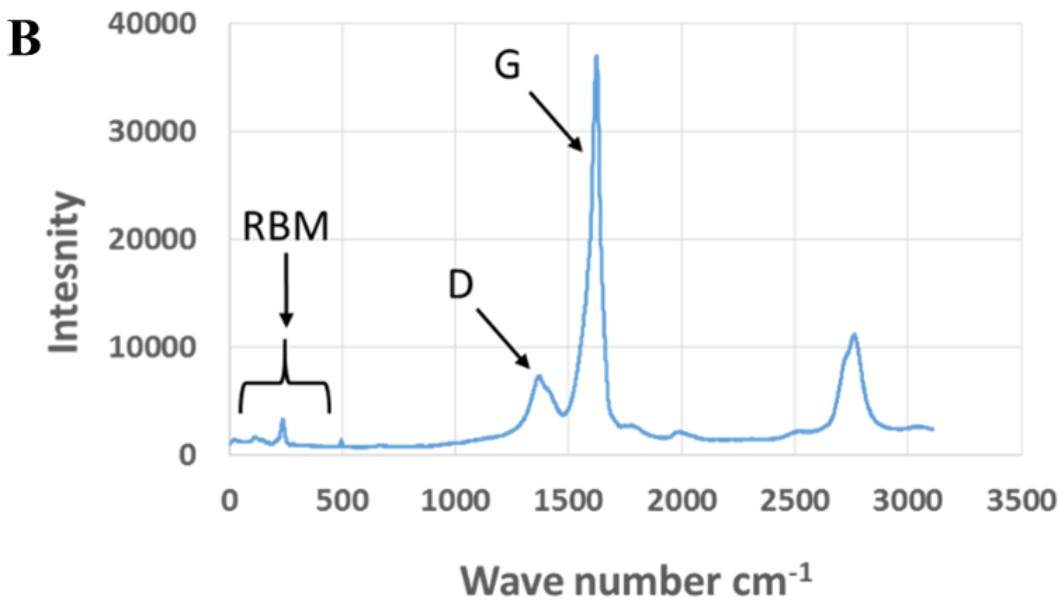

**Figure 2** **Comparison of Raman spectra of pristine and aged CNTRENE C100LM CNT material.** (A) Pristine CNTRENE. (B) CNTRENE artificially aged in a QUV Accelerated Weathering chamber for 12 days.

average doubling time of 29.6 ($\pm$0.9 min) upon exposure of up to 1.05 µg/ml CNTRENE material. These doubling times are similar to that of the untreated cells of 30.3 $\pm$ 0.5 min (Fig. 6C). Again, no correlation was seen between doubling time and treatment with increasing concentrations of pristine ($r = -0.1103$, $n = 9$ for each concentration, $p = 0.5490$) or aged ($r = 0.2342$, $n = 9$ for each concentration, $p = 0.1896$) CNTRENE material.

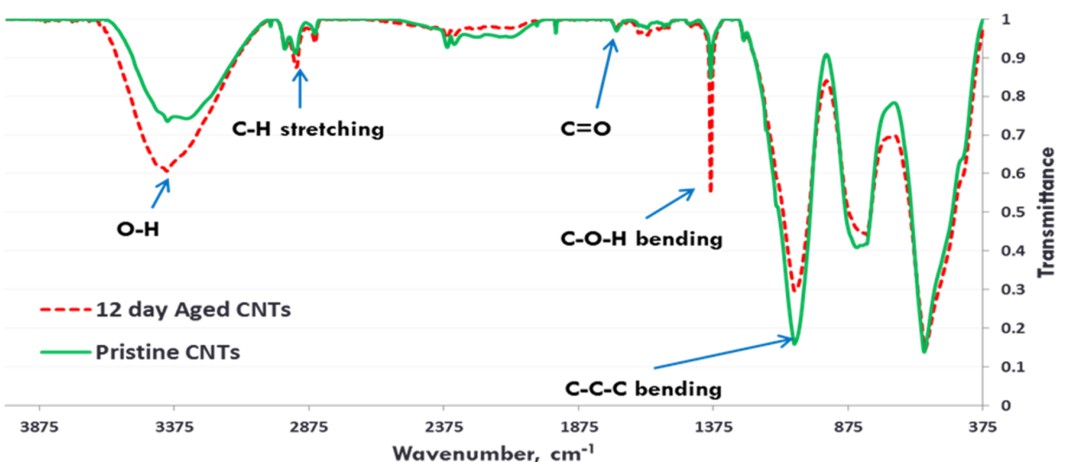

**Figure 3** FTIR Spectra of Pristine (green, solid) and aged (red, dotted) CNTRENE CL100LM CNT material.

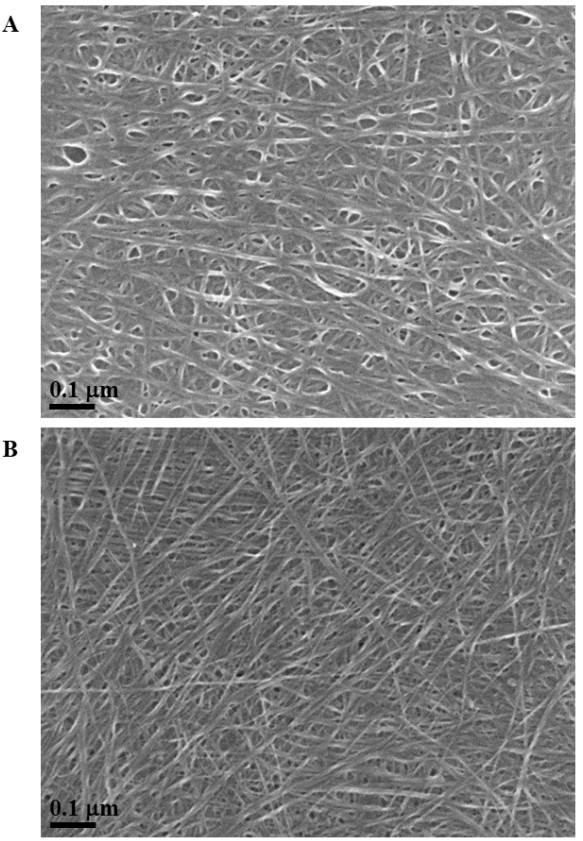

**Figure 4** SEM of aged and pristine CNTRENE C100LM CNT material. (A) CNTRENE artificially aged in a QUV Accelerated Weathering chamber for 12 days. (B) Pristine CNTRENE Images from JEOL JSM-7600F field emission SEM under vacuum ($9.6 \times 10^{-5}$ Pascal) with accelerating voltage of 5.0 kV (A) and 10.0 kV (B). Total magnification was 100,000×.
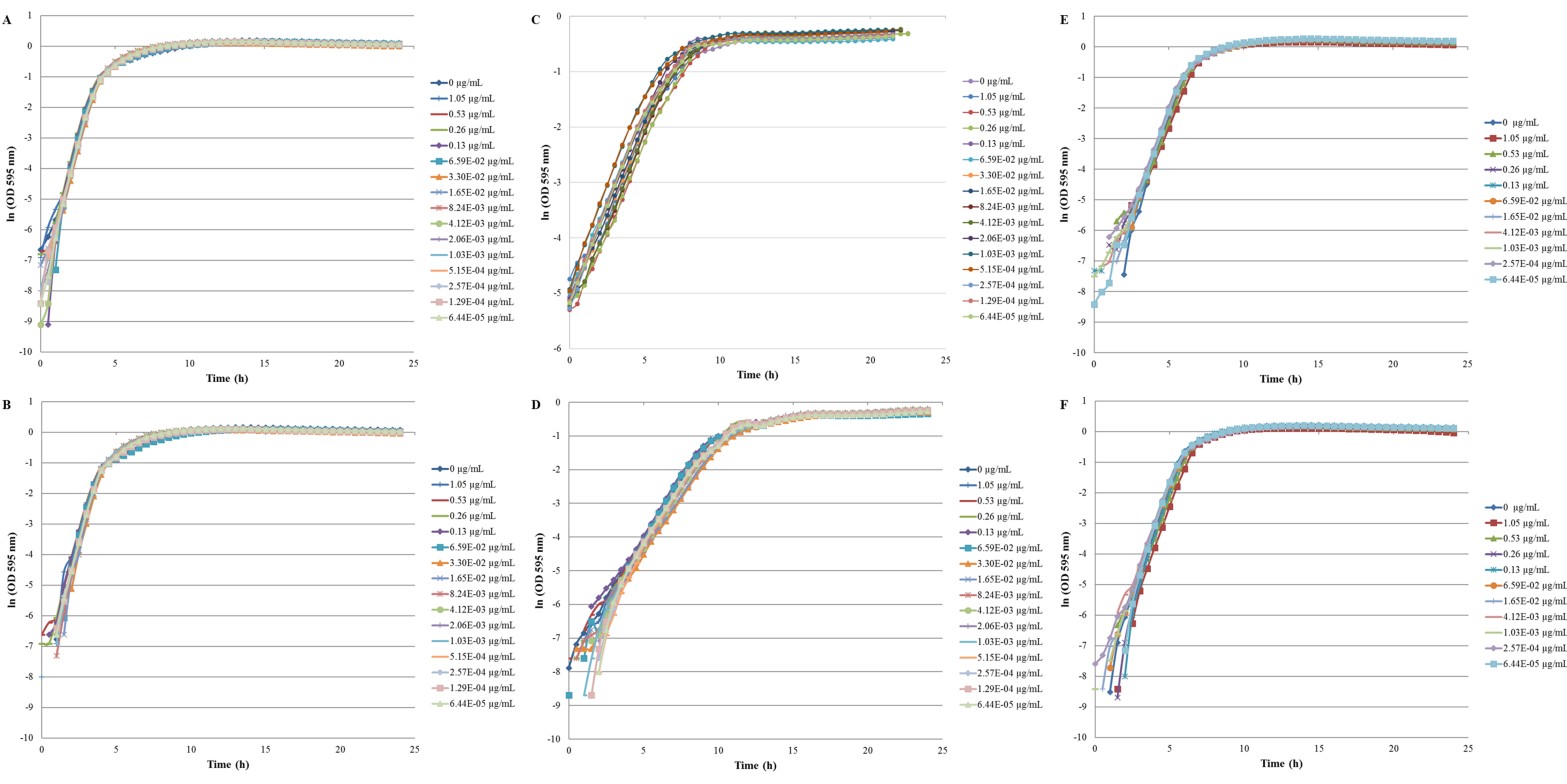

**Figure 5** **Comparison of *E. coli.* growth when exposed to pristine or aged CNTRENE C100LM CNT material.** Controls cells were grown in medium without CNT treatment. *E. coli* K12 were grown in the presence of 0–1.05 μg/ml pristine or aged CNTRENE C100LM. (A) *E. coli* exposed to pristine CNTRENE C100LM and grown in LB at a pH of 7. (B) *E. coli* exposed to aged CNTRENE C100LM and grown in LB at pH 7. (C) *E. coli* exposed to pristine CNTRENE C100LM and grown in M9 medium at a pH of 7. (D) *E. coli* exposed to aged CNTRENE C100LM and grown in M9 medium at pH 7. (E) *E. coli* exposed to pristine CNTRENE C100LM and grown in LB at a pH of 5. (F) *E. coli* exposed to aged CNTRENE C100LM and grown in LB at pH 5. Error bars have been omitted for clarity.

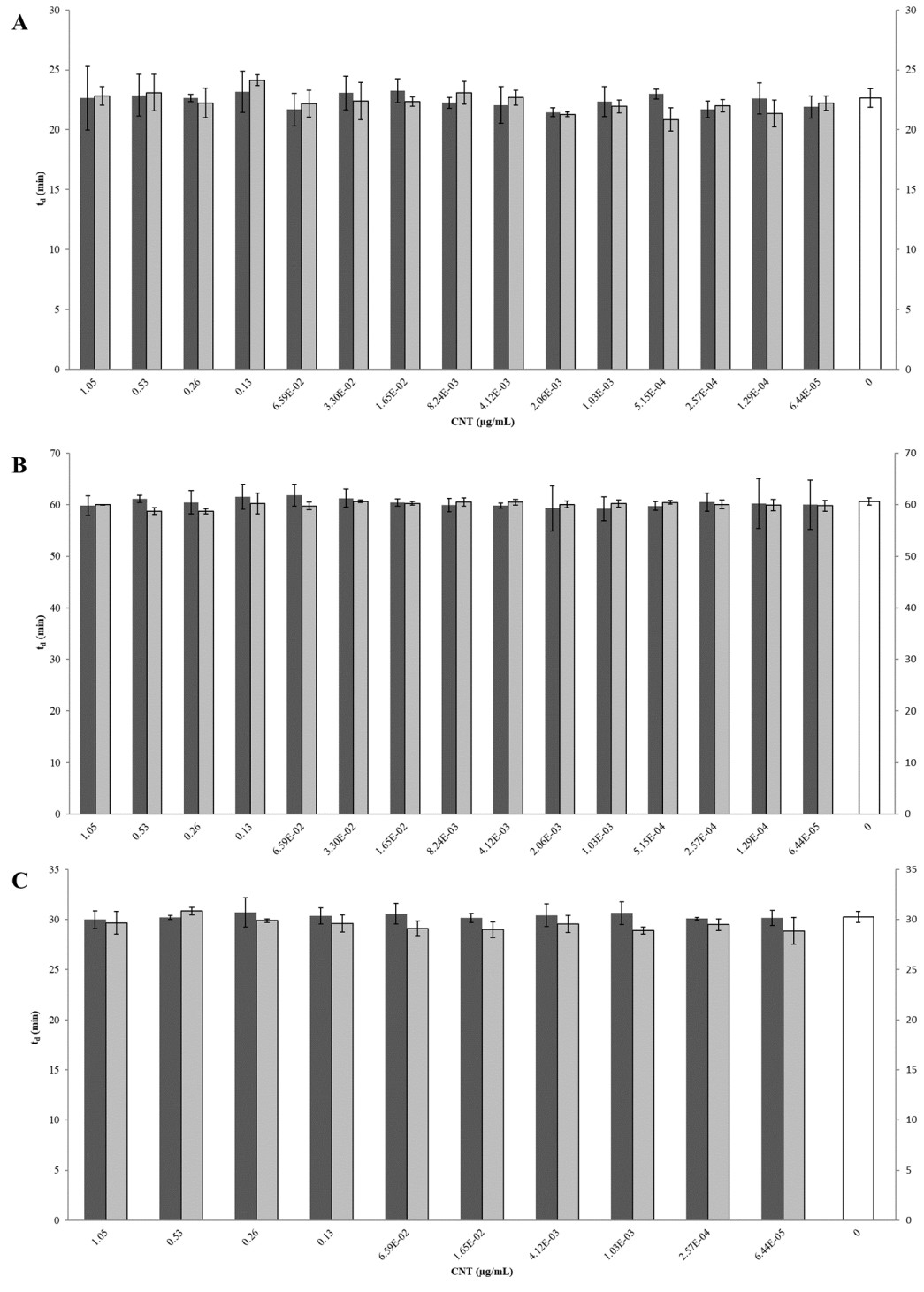

**Figure 6** **Comparison of doubling times ($t_d$) of *E. coli* K12 exposed to pristine or aged CNTRENE C100LM CNT material.** Treatment with either pristine (dark gray) or artificially aged (light gray) CNTRENE C100LM. Controls (unfilled bar) were grown in medium without CNTRENE C100LM. (A) *E. coli* grown in LB at pH 7. For pristine CNTRENE C100LM treatment $t_d$ ranged from 20.3 min–25.3 min with an average of 22. 4 min ($\pm 1.2$ min). 

## Antibacterial plate counts

Significant optical interference was observed at CNTRENE material concentrations over 1.05 $\mu$g/ml, making MIC microtiter assays infeasible for the assessment of cytotoxic effects. Therefore, growth effects of pristine CNTRENE material at concentrations higher than 1.05 $\mu$g/mL on *E. coli* K12 were evaluated by a modified spot plate technique (see Methods). The CFU/mL were calculated after 24 h of exposure to pristine CNTRENE material at final concentrations of 0 $\mu$g/ml (control), 8.44 $\mu$g/ml, 16.88 $\mu$g/ml, and 33.75 $\mu$g/ml. The calculated CFU/mL were $1.47 \times 10^9 \pm 7.87 \times 10^8$ CFU/ml ($n = 9$), $1.24 \times 10^9 \pm 9.26 \times 10^8$ CFU/ml ($n = 9$), $1.20 \times 10^9 \pm 6.24 \times 10^8$ CFU/ml ($n = 9$), and $1.87 \times 10^9 \pm 5.00 \times 10^8$ CFU/ml ($n = 9$), respectively (Fig. 7). These data were found to be Gaussian by a D'Agostino-Pearson normality test, and the differences between groups were not significant as determined by one-way ANOVA ($p = 0.2138$). Taken together with the MIC assays described above, these data imply that CNTRENE material exposure up to 33.75 $\mu$g/ml does not significantly impact the growth of *E. coli*.

## Electron microscopy imagining

Morphological changes of *E. coli* K12 grown in LB pH 7 and exposed to pristine CNTRENE material at concentrations at and above 1.05 $\mu$g/mL were evaluated by electron microscopy. With SEM, control cells visualized at 10,000$\times$ and 25,000$\times$ appeared as morphologically normal bacilli, with intact outer membranes and lengths ranging between 1 $\mu$m and 2 $\mu$m (Figs. 8A–8B). After 24 h exposure to pristine CNTRENE material (33.75 $\mu$g/mL and 16.88 $\mu$g/mL), cells had similar morphologies to control samples were within the typical 1 $\mu$m to 2 $\mu$m length range of *E. coli* (Figs. 8C–8F).

As was observed with SEM, AFM images of control *E. coli* K12 cells appeared as morphologically normal bacilli with intact outer membranes, lengths between 1 $\mu$m to 2 $\mu$m, and diameters of 0.5 $\mu$m (Fig. 9A). After 24 h exposure to pristine CNTRENE material (33.75 $\mu$g/mL, 16.88 $\mu$g/mL, and 1.05 $\mu$g/mL), cells had normal morphological features including cell length (1 $\mu$m–2 $\mu$m) and diameter (0.5 $\mu$m), similar to control cells (Figs. 9B–9D).

In SEM images, CNTRENE material was primarily observed to be adjacent to bacterial cells and not in direct contact with the cell surface. Although, some CNTRENE material was in direct contact with outer membranes of the cells, no damage to outer membranes, such as physical puncturing, was observed. Due to the resolution limitations of the instrument,

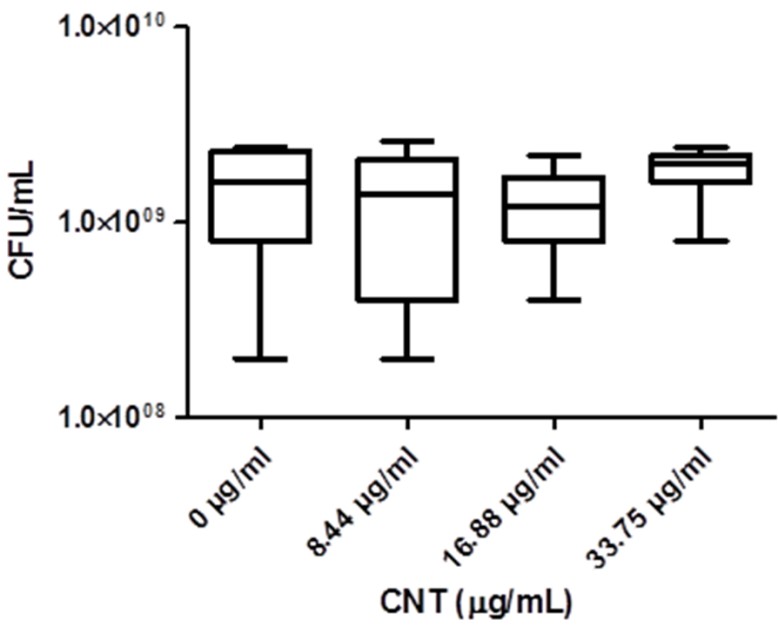

**Figure 7** **Box-and-Whisker of antibacterial plate counts.** CNTRENE C100LM exposure in *E. coli* K12 grown in LB pH 7. Data were determined to be Gaussian by a D'Agostino-Pearson normality test. Differences were not significant between groups as determined by one-way ANOVA ($p = 0.2138$, $n = 9$).

no CNTRENE material structures could be positively identified in AFM images, so no association between cells and CNTRENE material was directly observable. However, cells appeared intact, without abnormalities in cell morphology, which corresponds to the SEM images. Taken together, these data suggest that this CNTRENE material does not physically damage *E. coli*, which is in agreement with the data demonstrating normal growth upon CNTRENE material exposure.

## RNA sequencing

Global transcriptional changes were examined by RNA sequencing to examine if any regulatory changes were responsible for the tolerance of *E. coli* to CNTRENE material. *E. coli* K12 cultures were grown to mid-log phase in M9 medium containing 1.05 μg/ml CNTs and compared to cultures without CNTRENE material exposure. Control samples had an average of 2,199,975 reads (6,599,925 total reads) and experimental samples had an average of 2,592,180 reads (7,776,541 total reads). All control and experimental samples had sequence lengths of 35–51 bp with a GC content of approximately 54%, closely mirroring the genomic GC percentage of 50.8% (*Riley et al., 2006*). Gene expression of *E. coli* K12 exposed to pristine CNTRENE material was compared to native gene expression with the *E. coli* K12 MG1655 reference genome used for mapping sequencing reads. Of the 4464 open reading frames (ORFs) (NCBI accession NC_000913), 4,314 genes were mapped indicating that 96.6% of all genes were expressed. Of the 4,314 genes mapped, 186 genes were differentially expressed using a 2-fold change between control and experimental samples as a threshold. Of these 186 genes, 26 genes were upregulated in the experimental CNT exposed samples, and 160 genes were downregulated (Fig. S1). However, only three genes (*pptA*, *alpA*,

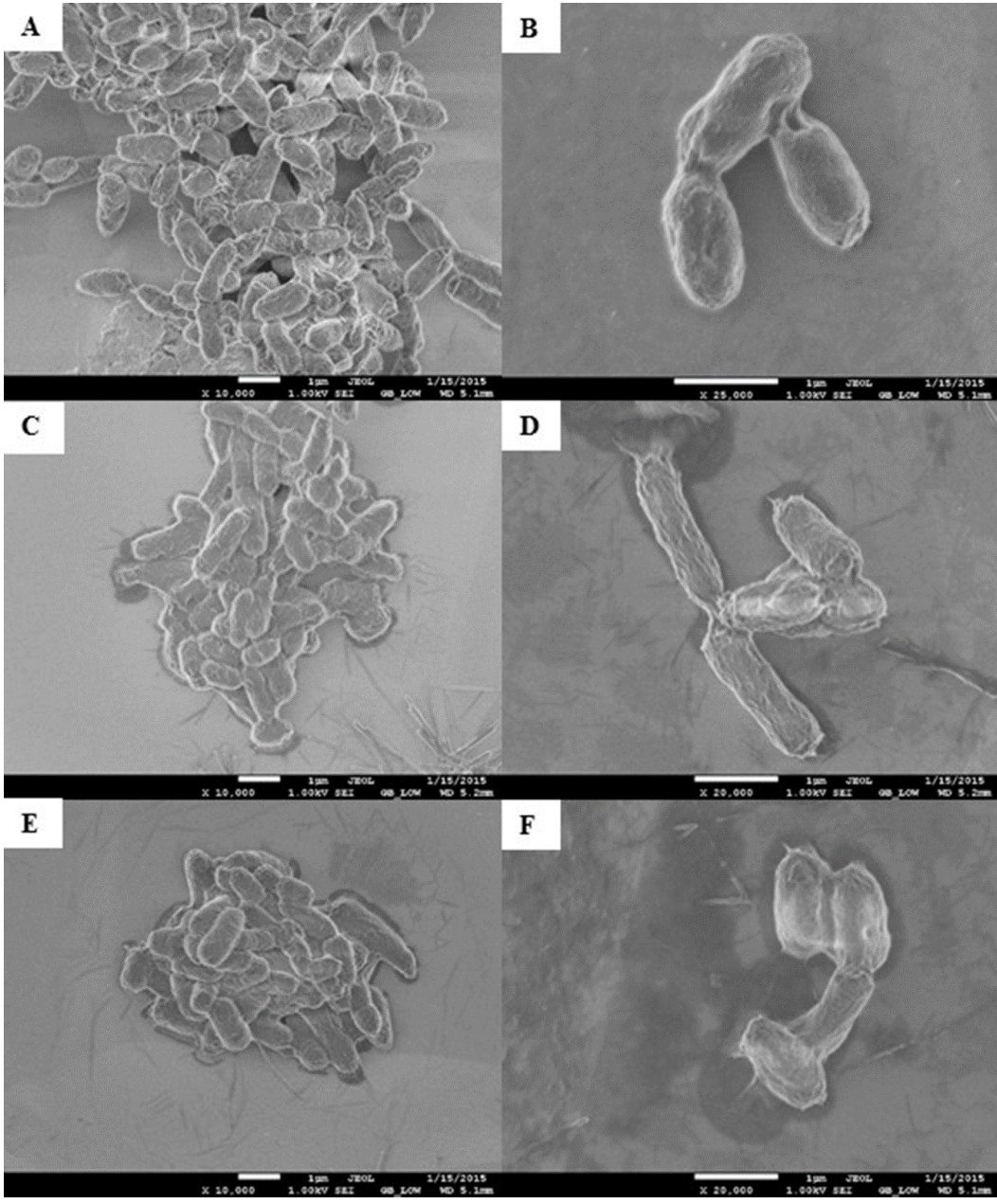

**Figure 8** **Scanning electron microscopy (SEM) images of *E. coli* K12 after 24 h exposure to pristine CNTRENE C100LM.** Images from JEOL JSM-7600F field emission SEM under vacuum ($9.6 \times 10^{-5}$ Pascal) with accelerating voltage of 1.00 kV. The scale bar is 1 $\mu$m. (A–B) 0 $\mu$g/mL (control), (C–D) 16.88 $\mu$g/mL, (E–F) 33.75 $\mu$g/mL pristine CNT exposure. Total magnification was 10,000$\times$ (A, C, E), 25,000$\times$ (B), and 20,000$\times$ (D, F).

and *mgtL*) were expressed at a significantly different level in the experimental samples after correcting for false discovery rate of 0.05 using the Benjamini–Hochberg method (*Benjamini & Hochberg, 1995*). The *pptA* and *alpA* genes were considered significantly downregulated with a 2.5-fold change ($p = 0.0272$) and 35.1-fold change ($p = 0.0227$), respectively. The *mgtL* gene was the only gene to be upregulated, with an 85.3-fold increase

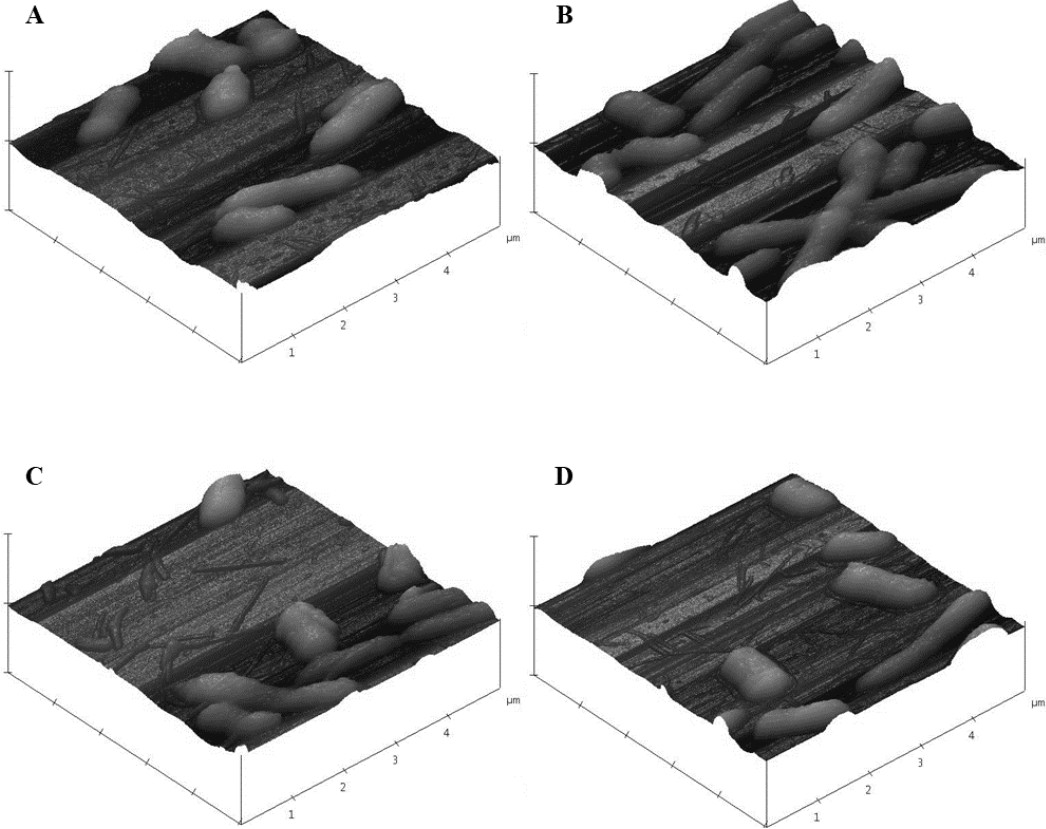

**Figure 9** **Atomic force microscopy (AFM) images of *E. coli* K12 after 24 h exposure to pristine CN-TRENE C100LM.** Three dimensional images from Veeco Dimension 3100 with a Nanoscope IIIA Controller using tapping mode and a silicon tip (radius of 8.0 nm) under atmospheric conditions. All images captured with a scan rate of 1.001 Hz and 512 samples. Data scale for all images was 2.000 μm with X position of −19783.4 μm and Y position of −42151.3 μm. (A) 0 μg/mL (control), (B) 1.05 μg/mL, (C) 16.88 μg/mL, (D) 33.75 μg/mL pristine CNTRENE C100LM exposure.

in expression in experimental samples ($p = 1.87 \times 10^{-7}$). The *pptA* gene (COG 1942) is predicted to encode a 4-oxalocrotonate tautomerase that functions in degradation pathways for xenobiotic aromatic compounds (*Kanehisa & Goto, 2000*). The *alpA* gene (COG 3311) is a prophage regulatory protein that is part of a group of DNA transcription regulators within the MerR superfamily, and regulators within this family have been shown to regulate in response to environmental stressors, such as heavy metals (*Marchler-Bauer et al., 2015*). However, the genes downstream of this ORF encode proteins associated with the cryptic prophage CP4-57 and are not known to be associated with stress responses. The *mgtL* gene acts as a leader sequence to the downstream *mgtA* gene (COG 0474), which is a $Mg^{2+}$ transport protein (*Park et al., 2010*). The *mgtL* leader sequence serves a riboswitch for the $Mg^{2+}$ porin, which allows expression of the porin to be regulated by the availability of proline and $Mg^{2+}$ (*Park et al., 2010*). Regardless, the role of all three genes identified by RNA-seq in response to CNTRENE material exposure is unclear as all three genes have dissimilar function and are not part of a general stress response that would be expected

from known mechanisms of CNT toxicity, such as physical interaction (e.g., cell envelope damage), ROS generation, or oxidative stress (*Nel et al., 2006*; *Kang et al., 2008*).

## DISCUSSION

With the emergence of nanotechnology and the growing number of applications for CNTs, such as biosensors (*Wang, Musameh & Lin, 2003*; *Chen et al., 2003*; *Huang et al., 2004*; *Trojanowicz, 2006*; *Timur et al., 2007*; *Yun et al., 2007*) and vaccine/drug delivery systems (*Kam et al., 2005*; *Cai et al., 2005*; *Bianco, Kostarelos & Prato, 2005*; *Liu et al., 2007*), it is important that the safety and potential impacts of nanoparticles on environmental communities are fully evaluated. The potential cytotoxic effects on microbial communities is an important consideration during a chemical life cycle analysis, as microorganisms play a vital role in environmental nutrient cycling, are essential for the maintenance of animal life, and play a role in health and disease. Disruption of the microbiota of an ecosystem has wide reaching consequences. However, evidence that exposure to realistic doses of nanomaterials causes acute toxicity is limited. Analysis of toxicity is also complicated by the lack of correlation between toxicity and nanoparticle size (*Valsami-Jones & Lynch, 2015*). Further complicating assessment of nanoparticle toxicity is the issue of aging (i.e., weathering upon environmental exposure), which can alter the physiochemistry and resulting toxicity of the material. This makes examining the effect of aging vital to the assessment of nanotoxicity.

In this study, the growth of *E. coli* K12 was not inhibited by pristine or aged CNTRENE material over the tested concentration range of up to 33.75 µg/ml under any growth conditions tested. Furthermore, no morphological changes were observed by SEM or AFM in which *E. coli* were exposed to CNTRENE material. It may be that the negatively charged carboxyl functional groups of CNTRENE material CNTs are partially repelled by the net negative charge of the bacterial cell, which could account for the lack of toxicity observed. This is in agreement with studies examining the cytotoxicity of fullerenes on bacteria. Fullerenes are a carbon nanomaterial consisting of a cage-like sphere of carbon boned in hexagonal or pentagonal arrangements, and CNTs can be considered a cylindrical form of a fullerene with similar surface and atomic structure and composition. In fullerenes, cationic functionalization (e.g., $-NH_3$) is generally associated with increased cytotoxicity compared to anionic functionalization (e.g., $-COOH$) (*Jensen, Wilson & Schuster, 1996*; *Bosi et al., 2003*; *Oberdörster, Oberdörster & Oberdörster, 2005*). Regardless of the mechanism, these data suggest that CNTRENE material exposure is benign to *E. coli*.

Despite the lack of physical alteration or growth effect of CNTRENE material exposure, three genes (*pptA*, *alpA*, and *mgtL*) were identified as differentially regulated in CNTRENE material-exposed cells. However, the role of these three genes is enigmatic because of their unrelated cellular roles and lack of a link to known stress response pathways associated with CNT exposure (*Nel et al., 2006*; *Kang et al., 2008*). Yet the lack of large changes in gene expression is not surprising given that growth was not significantly impacted by CNT exposure. It would be expected that gene regulation may play a role in normalizing growth behavior in CNTRENE material-exposed cells if the CNTs were at sub-toxic

concentrations. However, almost no gene regulation was seen and the three genes that were regulated were not differentially expressed to a large extent (i.e., >100-fold). Interestingly, CNT exposure has been previously reported to influence gene regulation in bacteria. For example, CNT exposure to non-functionalized single-walled CNTs and multi-walled CNTs was reported to activate genes associated with membrane and oxidative stress (*Kang et al., 2008*). Recently, exposure of *Paracoccus denitrificans* to carboxyl-functionalized single-walled CNTs (10- µg/ml–50 µg/ml) inhibited cell growth by reducing the expression of genes involved in DNA repair, glucose metabolism, and energy production (*Zheng et al., 2014*). The CNTRENE material used here has been recently associated in gene regulation in *Saccharomyces cerevisae* (*Woodman et al., 2016*). Here, they report 82 genes differentially expressed, of which 56 were up-regulated and 26 were down-regulated. Approximately 20% of the genes were implicated in increasing the rate of membrane transport, suggesting a detoxification route. This correlated to an observed increase in growth rate and decreased cell density of CNTRENE material-exposed cells. Yet none of these previously reported alterations in gene expression were observed in CNTRENE material-exposed *E. coli*. This may be due to disparate CNT toxicity mechanisms in bacteria and eukaryotes. For example, CNT toxicity in eukaryotes is often attributed to their uptake, especially by phagocytic cells, resulting in frustrated phagocytosis leading to the production of reactive oxygen species and release pro-inflammatory cytokines (*Brown et al., 2007*; *Johnston et al., 2010*). To our knowledge, bacteria are not known to uptake CNTs. Taken together, these data suggest that CNTRENE material exposure does not cause cell damage, death, or influence growth of *E. coli* K12, and that exposure to elevated levels up to 33.75 µg/ml CNTRENE material is innocuous to *E. coli* K12.

The lack of a deleterious effects CNTRENE material exposure is somewhat surprising given that many CNTs have been reported as having cytotoxic effects on bacteria, including *E. coli*. However, it should be addressed that there are many contradictory findings about the bacterial cytotoxicity of CNTs, which have been attributed to the variety of CNTs available including differences in purity and heavy metal content left over from CNT production (*Yang et al., 2010*). While some studies have found strong cytotoxic activity with carboxyl-functionalized CNTs (*Arias & Yang, 2009*), others have reported the opposite finding (*Lewinski, Colvin & Drezek, 2008*). Most studies examine cytotoxicity of CNTs that are artificially coated onto membranes by filtering. This procedure often is used to concentrate CNTs, likely providing exposure concentrations above those that would be obtained naturally. It also forces a CNT-cell interaction that may not accurately reflect planktonic bacterial cytotoxicity or cytotoxicity in biofilms. Moreover, many cytotoxicity studies that do report CNTs as highly toxic in planktonic cell cultures only report toxicity associated with the CNT-cell aggregates (*Kang et al., 2007*; *Kang et al., 2008*). This also artificially inflates toxicity because the majority of the cells are suspended and not in association with CNTs or CNT aggregates. For example, non-functionalized SWCNTs were reported to cause 80% loss of *E. coli* K12 viability in liquid cultures. However, this was only for CNT-bacterial aggregates, while the viability for cells in free suspension without CNT-association was only reduced by 8%. This was equivalent to the loss of viability of untreated cells (*Kang et al., 2008*). This suggests that a physical interaction is

necessary for CNT cytotoxicity. Most cells grown in liquid culture are planktonic and not CNT-associated. Consequently, studies that only examine these associations and ignore the majority of the cells (i.e., cells in bulk solution) likely greatly overestimate the cytotoxic effect of the CNT in question. Due to the variation in reported effects and the variety of potential applications, it is imperative that the effects of each distinctive type of CNT and their characterization is adequately evaluated using standardized methods to obtain a clear picture of toxicity.

In summary, we have shown that pristine and aged CNTRENE® C100LM CNT materials are not deleterious to the growth of *E. coli* at environmentally relevant concentrations up to 33.75 μg/ml. Furthermore, gene expression is not altered in a significant way, indicating that there is no need for *E. coli* to adapt to exposed conditions. It should be mentioned that other microorganisms may respond differently. Therefore, it may be useful to examine microbial communities, natural or artificial (e.g., microcosms), that are exposed to CNTRENE material to more completely understand its microbial cytotoxicity. This study highlights the importance of the continued toxicity screening of nanomaterials which is especially important in light of contradicting reports of nanomaterial cytotoxicity. Furthermore, there has been no physical or chemical parameter (e.g., functionalization) that has been shown to be predictive of cytotoxicity (*Valsami-Jones & Lynch, 2015*). Standard methods for nanoparticle cytotoxicity also do not appear to exist and must be developed. Current studies evaluating cytotoxic effects of CNTs on bacterial cells demonstrate the need for adequate characterization of the CNT materials tested, because physical and chemical properties of CNTs, including length, diameter, functionalization, and metal contamination have shown an effect on cell viability observed in bacterial cells. Furthermore, the degree of CNT dispersion in the solution can often impact toxicity, with dispersed CNTs being more toxic to bacterial cells than aggregated CNTs (*Kang et al., 2008*; *Liu et al., 2009*). For example, surfactants used to suspend single-walled CNTs also affect microbial cell viability and demonstrates that there are compounding factors when evaluating cytotoxicity of nanomaterials (*Dong, Henderson & Field, 2012*). Interpretation of data across labs will continue to be problematic until these standard practices are developed. Here we have described a set of simple standard assays that may be done to establish microbial cytotoxicity of nanomaterials.

## ACKNOWLEDGEMENTS

We would like to thank Brewer Science, Inc. for the generous gift of the CNTs; Rishi Patel for training and help gathering SEM and AFM data, and for providing information regarding physical characteristics of the CNT material; Dr. Wenping Qiu in the School of Agriculture for use of Agilent Technologies 2100 Bioanalyzer; and Angela Schmoldt and Raymond Hovey at the University of Wisconsin- Milwaukee for their assistance with RNA sequencing and bioinformatic analysis.

### Funding

This work was supported by the Missouri State University College of Natural and Applied Sciences through the Nanotech Fund. The funders had no role in study design, data collection and analysis, decision to publish, or preparation of the manuscript.

### Grant Disclosures

The following grant information was disclosed by the authors:
Missouri State University College of Natural and Applied Sciences through the Nanotech Fund.

### Competing Interests

The authors declare there are no competing interests.

### Author Contributions

- Brittany Twibell and Geoffrey Manani performed the experiments, analyzed the data, wrote the paper, prepared figures and/or tables, reviewed drafts of the paper.
- Kalie Somerville analyzed the data, wrote the paper, prepared figures and/or tables, reviewed drafts of the paper.
- Molly Duszynski performed the experiments, analyzed the data, reviewed drafts of the paper.
- Adam Wanekaya and Paul Schweiger conceived and designed the experiments, analyzed the data, contributed reagents/materials/analysis tools, wrote the paper, prepared figures and/or tables, reviewed drafts of the paper.

### Data Availability

   GEO Accession GSE99042.

### Supplemental Information

Supplemental information for this article can be found online at http://dx.doi.org/10.7717/peerj.3721#supplemental-information.

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
