# Peer review of "Influence of CNTRENE® C100LM carbon nanotube material on the growth and regulation of Escherichia coli"

_PeerJ, doi:10.7717/peerj.3721_

## Round 0.1 · original submission · Minor Revisions

Please, include in the revised version of your MS all the suggestions and criticism mentioned by the three reviewers. However, a bit more work is required than only an editorial Minor Revision. On the other hand, I do not see the absolute necessity for additional experimentation.

Reviewer 1 ·

Basic reporting

The authors have investigated how exposure of carboxylated nanotubes to basic parameters associated with aging affect an Escherichia coli strain mechanically with regard to cell morphology, as well as cultural vialbility, batch culture growth kinetics, and regulation of presumptively pertinent stress response genes.

The paper is very well written, organized in a clear and cogent manner, and the data are presented in well-designed and easy to understand figures. However, the manuscript is lengthy given the purpose of the study and the background information necessary to both understand why the work was carried out and to interpret the results. Much of the introductory material would not be necessary for scientists familiar with the field who are most likely going to use the paper. It reads much like a "mini-review" and could be truncated to more succinctly focus on the most pertinent background literature.

Experimental design

The experimental design is well thought out and clearly described. Given that the overall finding is that aging of the material being examined exerted no detectable deleterious effects on the model organism, the work would have been strengthened had other organisms been included. For example, do gram-positive or other phylogenetically disparate organisms react differently to the same exposures? Aging clearly affected the structure of the materials, therefore the absence of an effect on only one organism begs the question is this true of bacteria in general, or is it organism dependent?

Validity of the findings

The data as presented in the figures and summarized in the Results cogently support the conclusions reached. There is no obvious overreach, and the data were analyzed for statistical significance.

Additional comments

This work has value in that it emphases the facts that mixed results predominate in the existing literature, and it suggests mechanisms by which work can be more standardized to potentiate efforts to identify why. It is a well thought out and exhaustive study which is a credit to the senior author and his co-workers.

Specific Comments:

Line 57 and thruout-change "since" to "because".
Line 60- define "SNUR".
Lines 76-77- Because no studies of this particular functional group have been conducted, it would have been especially helpful to of at least included a gram-positive bacterium for comparison.
Line 91-94- Highly redundant background material.
Lines 115-116- Details lacking regarding how the spectroscopy was conducted.
Line 124- What is the source for the software?
Line 181- Change "are" to "is".
Lines 375-376- It is not clear how the authors would be able to know this.

·

Basic reporting

No comment

Experimental design

Ln 102-108: was the physical characterization performed by the authors or reported by the manufacturer? Please completely describe the CNTs as completely as possible. Use of the SEM micrographs is one method that could make use of existing data to more fully characterize the nanomaterials.

Please describe the CNT dispersion method so that it can be exactly replicated by others. Did single particles result? Were they tangled? Were they well dispersed?

Ln 191: The authors state that their aging technique produced CNTs with “morphological and functional changes as the CNTs are aged in conditions that mimic prolonged environmental exposure.” However, there is no information on what their material is compared to as a reference for aging. There is a single citation in the paragraph that discusses the general concept of environmental aging but, I don’t believe, has any specific information. The authors are asked to expand on this to help the reader understand the relevance of their aging technique.

Validity of the findings

No comment

Additional comments

This manuscript describes the effects of carbon nanotubes on the short-term growth of E. coli K12. The results showed no effects at any of the concentrations tested by any of 3 different evaluations: growth rate, morphology and gene expression. The study was well done and appropriately described. The results are clear and the interpretation consistent with the data presented. The discussion is concise and pertinent to the work. Overall, this work contributes to the literature in better understanding the effects of CNTs on living systems. Several minor points are in the Experimental Design section.

Reviewer 3 ·

Basic reporting

no comment

Experimental design

This reviewer wonders if E. coli represent a good model for the toxicity testing of CNTs. It is widely accepted that toxicity of CNTs is generally induced by the presence of contaminants (e.g. arsenic or other heavy metals carried over from the synthesis production) or to alternative mechanisms that involve the uptake of CNTs by the cellular system (e.g. frustrated phagocytosis in humans). In this respect, this reviewer is not aware that active uptake of CNTs by E. coli is possible or if has ever been reported. Please cover this part in the introduction and in the discussion.

The characterization of the CNTs does not look complete. As explained in the previous comment, often the toxicity of CNTs is linked to the presence of contaminants, which, however, were not determined for the CNTs used in this study.

How do the CNTs physico-chemical properties change after aging? Do they change in size (e.g. becoming shorter?). The characterization of the CNTs should be performed after aging process and compared to those obtained in pristine conditions. Also, some TEM or SEM pictures would help to understand what happened to the CNTs during the aging process. In the absence of this data the statement at line 191-193 can only be considered an educated guess.

The medium in which the CNTs were dissolved during the accelerated aging process is not clear to this reviewer. Were the CNTs dispersed in distilled water? Were they in dry state? This reviewer thinks that a better choice would have been to disperse the CNTs in the medium that was used for the growing and exposure of E. coli.
It is the opinion of this reviewer that an aging process performed in dry conditions or in distilled water cannot be considered as “environmental aging”, but it should be rather considered as “shelf aging”, since the environmental conditions to which the CNTs will be exposed are not dry state or water dispersion. It should be considered that the presence of salts, proteins or other organic molecules (as in the M9 or LB medium) will definitely modify the behaviour of the CNTs, due to the formation of the bio-corona and that this process will definitely change after aging.

Validity of the findings

This reviewer finds difficult to compare the observed results with previous studies on S. cerevisiae. In one case we have bacteria, while in the other case we have eukaryotic cells. Not only most of the cell machinery systems are different, and regulated in a different way, but also the respective dimension of the cell is different (roughly factor 10), which might have a strong impact on the uptake of CNTs by the different cells.

It is also different to compare this study to previous studies in the absence of proper and extensive characterization of the CNTs, which should include the presence and quantification of impurities. Were the CNTs and CNTRENE used by Zheng and by Woodman, respectively, comparable to those used in this study? Without this information it is not possible to compare the different studies.

Additional comments

In the manuscript entitled “Influence of CNTRENE CM100LM carbon nanotube material on the growth and regulation of Escherichia coli” by Twibell et al. the authors investigated the effects of pristine and aged carboxylate CNTs on the bacteria E. coli.

The topic addressed by the authors is absolutely interesting. The paper is well written and the data look reasonable. However, this reviewer thinks that the paper will definitely benefit from additional experiments, particularly in the characterization of the CNTs before and after accelerated aging.

---

## Round 0.2 · accepted · Accept

All three reviewers accepted well the revisions you made. Congratulations!

Reviewer 1 ·

Basic reporting

Accept.

Experimental design

Accept.

Validity of the findings

Accept.

Additional comments

I feel that all issues I raised have now been satisfactorily addressed by the authors and that this manuscript in its present form exceeds the minimal criteria for publication. Thank you for the time and effort expended in improving your work.

·

Basic reporting

The revision addressed all reviewer requests.

Experimental design

The revision addressed all reviewer requests.

Validity of the findings

The revision addressed all reviewer requests.

Additional comments

The revision addressed all reviewer requests. The manuscript has been improved and provides a clear understanding of the experimental design, the results obtained and appropriate interpretation.

Reviewer 3 ·

Basic reporting

Nothing to add

Experimental design

Nothing to add

Validity of the findings

Nothing to add

Additional comments

This reviewer is satisfied with the corrections made to the manuscript.